# Variants of beta-glucan polysaccharides downregulate autoimmune inflammation

Cecilia Fahlquist-Hagert [1,3✉], Outi Sareila[1,2,4], Sofia Rosendahl[1,2] & Rikard Holmdahl [1,2✉]

Common infections and polysaccharides, from bacteria and yeasts, could trigger psoriasis and psoriatic arthritis (PsA), and possibly rheumatoid arthritis (RA). The objective of this study was to investigate the effects of β-glucan polysaccharides in the effector phase of arthritis and as regulators of psoriasis and PsA-like symptoms in mice. Collagen antibody induced arthritis was studied as a model of RA and mannan-induced psoriasis (MIP) was used as model for psoriasis and PsA, using mice with a mutation of Ncf1 on the B10.Q genetic background, making them highly disease susceptible. The mice were exposed to three common variants: 1,6-β-glucan, 1,3-β-glucan and 1,3-1,6-β-glucan. These β-glucans down-regulated disease in mice if administered simultaneously, before or after mannan. Interestingly, the protection was macrophage mannose receptor (MMR/CD206) dependent with a more pronounced protection long-term than short-term. The number of resident peritoneal macrophages decreased after in vivo challenge with β-glucan and mannan compared to mannan alone, whereas the numbers of infiltrating cells correspondingly increased, further indicating macrophages as key for β-glucan mediated regulation. At the doses tested, β-glucans could not induce arthritis, psoriasis or PsA in wild-type mice. However, β-glucans could ameliorate the PsA-like symptoms representing a new unforeseen possibility to explore for future clinical treatment.

[1] Medical Inflammation Research, MediCity Research Laboratory, University of Turku, FI-20520 Turku, Finland. [2] Medical Inflammation Research, Department of Biochemistry and Biophysics, Karolinska Institute, SE-17177 Stockholm, Sweden. [3]Present address: Aarhus University, Department of Biomedicine, Høegh-Guldbergs Gade 10, 8000 Aarhus, Denmark. [4]Present address: Department of Rheumatology and Inflammation Research, Institute of Medicine, Sahlgrenska Academy, University of Gothenburg, 40530 Gothenburg, Sweden. ✉email: cecilia.hagert@biomed.au.dk; rikard.holmdahl@ki.se

Psoriasis and psoriatic arthritis (PsA) are common diseases in humans, which are triggered by unknown environmental factors in genetically susceptible individuals causing chronic inflammatory disease. Psoriasis is characterized by inflammation and hyperproliferation in the skin and affects 2–3% of the population worldwide, and approximately 25% of the psoriasis patients will develop PsA phenotypes[1,2]. While the environmental triggers are unclear, physical injuries to the skin (the "Koebner response"), various infections, and inflammation-induced stimuli have been shown to initiate/exacerbate psoriatic lesions[3]. The natural polysaccharide mannan, from the baker's yeast *Saccharomyces cerevisiae*, has been shown to induce psoriasis and PsA-like disease (mannan-induced psoriasis, MIP) in mice[4]. PSORS1 is the major psoriasis susceptibility locus in humans[5]. In mice, MIP is regulated by Qa genes in the mouse PSORS1 locus and therefore mice with an H2q haplotype are more susceptible than the standard H2b haplotype[6]. In mice pre-injected with anti-collagen type II (aCol2) antibodies, binding joint cartilage, mannan triggers the development of chronic arthritis with similarities to rheumatoid arthritis (RA), a model denoted mannan enhanced collagen antibody-induced arthritis, mCAIA[4,7–9]. MIP and mCAIA are both driven by the innate immune system, with no critical dependency of adaptive immune cells such as B and T cells[4,7,10]. Both MIP and mCAIA are regulated by the NADPH oxidase type 2 (NOX2)-derived reactive oxygen species (ROS) as more severe disease develops in mice with a mutation in the neutrophil cytosolic factor 1 (*Ncf1*) gene[4,7,10]. The *NCF1* gene is polymorphic also in humans and is a major locus associated with autoimmune disease. Importantly, it is known that copy number variation of the *NCF1* gene is associated with rheumatoid arthritis and a *NCF1* single-nucleotide polymorphism represents the major genetic factor associated with systemic lupus erythematosus[11–13].

Humans are exposed to mannan through different sources as it is a component in yeasts, bacteria, and plants. Mannan often occurs together with the polysaccharide β-glucan[14]. Thus, when the immune response encounters mannan it often encounters β-glucan as well. β-glucans are widely occurring polysaccharides consisting of d-glucose monomers linked by β-glycosidic bonds, functioning as structural components of algae, bacteria, fungi, and plants. β-glucans have a defined structure–activity relationship but has been shown to have a varied biological activity depending on the source and structure of the β-glucan[15]. Indeed, in mice with a mutation in the ZAP70 gene, enhancing the activation of autoreactive T cells (SKG mice), 1,3-β-glucans (Curdlan) have been shown to induce arthritis[16] and spondylarthritis[17]. The development of arthritis by Curdlan is believed to operate through stimulation of Dectin-1[18]. Curdlan has also been shown to activate macrophages leading to a hyperinflammatory status in mouse models[19]. β-glucans have been shown to bind to Dectin-1, primarily expressed on monocytes/macrophages and neutrophils[20], CD14 (*Fcer2a*)[21], present primarily on macrophages but also on neutrophils and dendritic cells, and CD11b/CD18 (also known as complement receptor 3 (CR3)), present on neutrophils[22]. Anti-inflammatory effects of β-glucans have previously been described[23], for example in an in vivo model of colitis[24,25] were 1,3-β-glucan from mushroom (Lentinan) or fractions from β-glucan extracted from *C. albicans*, respectively, was able to protect from dextran sulfate sodium (DSS)-induced colitis mice with severe weight loss due to shortening and severe damage to the intestine. Furthermore, β-glucans have been shown to protect against ear edema induced by croton oil[26]. Humans are exposed to different β-glucan configurations on a regular basis and different configurations could affect the immune system differently[15–17,27]. In this paper, we investigate whether β-glucans have a protective or pathogenic effect on the development of psoriasis and psoriatic arthritis in the MIP model, focusing on the three common variants

of β-glucans; baker's yeast glucan (a 1,3–1,6 β-glucan), curdlan, a 1,3-β-glucan and pustulan, a 1,6-β-glucan.

## Results

### Selection and characterization of β-glucans.
To investigate whether beta-glucans can induce or protect against arthritis and psoriasis we selected three different β-glucans; 1,3–1,6 β-glucan from baker's yeast, curdlan (1,3-β-glucan) and pustulan (1.6-β-glucan). First, the solubility and purity were investigated, to exclude possible effects of contaminations. It is important to note that all β-glucans are insoluble in water but can form suspensions with small precipitations, gelatinous flakes in white almost see-through color that at low β-glucan concentrations is injectable. To further confirm that this had no effect on our treatment design, small trials were performed where the β-glucan suspensions were compared to filtered β-glucan suspensions. The effect upon MIP was milder when the suspension had been filtered to remove the precipitations (Supplementary Figs. 1 and 2), highlighting the importance of using these compounds as suspensions to administer the intended dose and all (e.g., soluble and insoluble) forms of the product.

The presence of bacterial contamination in 1,3- and 1,6-β-glucans had been assessed by the manufacturer using HEK-Blue™ TLR2 and HEK-Blue™ TLR4 cells (Table 1). To further characterize the β-glucans, we performed a Limulus Amebocyte Lysate (LAL) assay to study possible endotoxin contamination in the β-glucans. Since some β-glucans are known to interfere with the LAL assay[28,29], we tested the β-glucans in the presence of β-G-Blocker to render the test more specific to endotoxin. Endotoxin was not detected in 1,3–1,6-β-glucan (Supplementary Fig. 3). Both 1,3- and 1,6-β-glucan resulted in a signal that was dose-dependently diminished by the 1,6-β-G-Blocker. In the 9:1 (v/v) excess of the blocker, the reading corresponded to endotoxin activity of 8 EU/mg of 1,3-β-glucan and 16 EU/mg of 1,6-β-glucan. We conclude that there is no endotoxin contamination of β-glucan that could affect the results. Regarding curdlan and pustulan the small possible contaminations are unlikely to affect both a disease-promoting and a possible disease protective effect. It should be seen in relation to the use of LPS, mannan, or lipomannan for enhancement of the classical CAIA method in which a far larger dose is needed to enhance arthritis and no protective effect can be noted[30].

### β-glucan from *S. cerevisiae* cannot induce arthritis.
As β-glucans have previously been shown to induce an inflammatory response[18] we addressed the possibility that β-glucans could induce arthritis, in similarity with mannan[4,7,10]. As a representative of 1,3–1,6-β-glucan, we chose to use glucan from *Saccharomyces cerevisiae* to investigate its effect on psoriasis induced by mannan, another polysaccharide from *S. cerevisiae*. We have earlier seen enhancement of CAIA with immunostimulants from bacteria[7,30]. Others have reported enhancement of autoimmune arthritis in mice by *Candida albicans* β-glucan[31,32], and thus we investigated whether the *S. cerevisiae* β-glucan also enhances CAIA. The dose of 800 μg was chosen because it is a high enough dose to be expected to boost CAIA[30,33]. To that end, 1,3–1,6-β-glucan was injected with or without prior exposure to anti-Col2 antibodies to investigate its ability to induce PsA or enhance RA-like symptoms in CAIA, similar to mannan[7] in a dose corresponding to previously successful immunostimulants[30]. However, 1,3–1,6-β-glucan alone did not induce any disease and did in combination with aCol2 antibodies cause only a mild transient disease (Fig. 1a, b), similar to what is normally seen with aCol2 antibodies alone[7]. The lack of arthritis is in line with Kelkka et al.[30] showing that zymosan, a β-glucan containing bacterial

**Table 1 Description and characteristics of the polysaccharides used in the study.**

| Name | Mannan | 1,3-1,6-β-glucan | 1,3-β-glucan | 1,6-β-glucan |
|---|---|---|---|---|
| Aliases | NA | β-1,3/1,6-glucan | Curdlan, β-1,3-glucan hydrate | Pustulan, β(1 → 6)-glucan |
| Trade name | Mannan from Saccharomyces cerevisiae—prepared by alkaline extraction | Glucan from baker's yeast (S. cerevisiae) | Beta-1,3-glucan from Alcaligenes faecalis | Dectin-1 agonist - beta-glucan from Lasallia pustulata |
| Catalog number[a] | M7504 | G5011 | tlrl-curd | tlrl-pst |
| CAS number[a] | 9036-88-8 | 9012-72-0 | 54724-00-4 | 37331-28-5 |
| Manufacturer | Sigma-Aldrich | Sigma-Aldrich | InvivoGen | InvivoGen |
| Molecular weight | 34.0–62.5 kDa[50] | 35–5000 kDa[51] | High molecular weight[a] Usually 66–680 kDa[52] | 20 kDa[a] |
| Polymer structure | Branched | Branched | Linear | Linear |
| O-glycosidic linkages in glucans | NA | 1 → 3 and 1 → 6 | 1 → 3 | 1 → 6 |
| Purity[a] | ≥99.9% | ≥98.00% | Not determined | Not determined |
| Solubility in water[a] | ca.50 g/l | 9.80–10.20 mg/ml | Insoluble | Insoluble |
| Solubility (turbidity)[a] | Clear to slightly hazy | Turbid with insolubles | Non-homogenous suspension with gelatinous precipitates | Non-homogenous suspension with gelatinous precipitates |
| Preparation for in vivo administration | Dissolved in PBS 10–20 mg/ml | Suspended in PBS ≤5 mg/ml | Suspended in PBS ≤ 3 mg/ml | Suspended in PBS ≤3 mg/ml |
| Additional Notes | Impurity (by TLC) ≤ 0.1%[a] | NA | The presence of bacterial contamination (e.g., lipoproteins and endotoxins) has been assessed using HEK-Blue™ TLR2 and HEK-Blue™ TLR4 cells. Curdlan induces TLR2 and TLR4 activity when used at concentrations higher than 100 μg/ml[a] | The presence of bacterial contamination (e.g., lipoproteins and endotoxins) has been assessed using HEK-Blue™ TLR2 and HEK-Blue™ TLR4 cells. Latest batch activated TLR2 at 100 μg/ml and TLR4 at 30 μg/ml and 10 μg/ml resulted in only weak activation of both.[a] |

NA not applicable.
[a]information from the manufacturer.

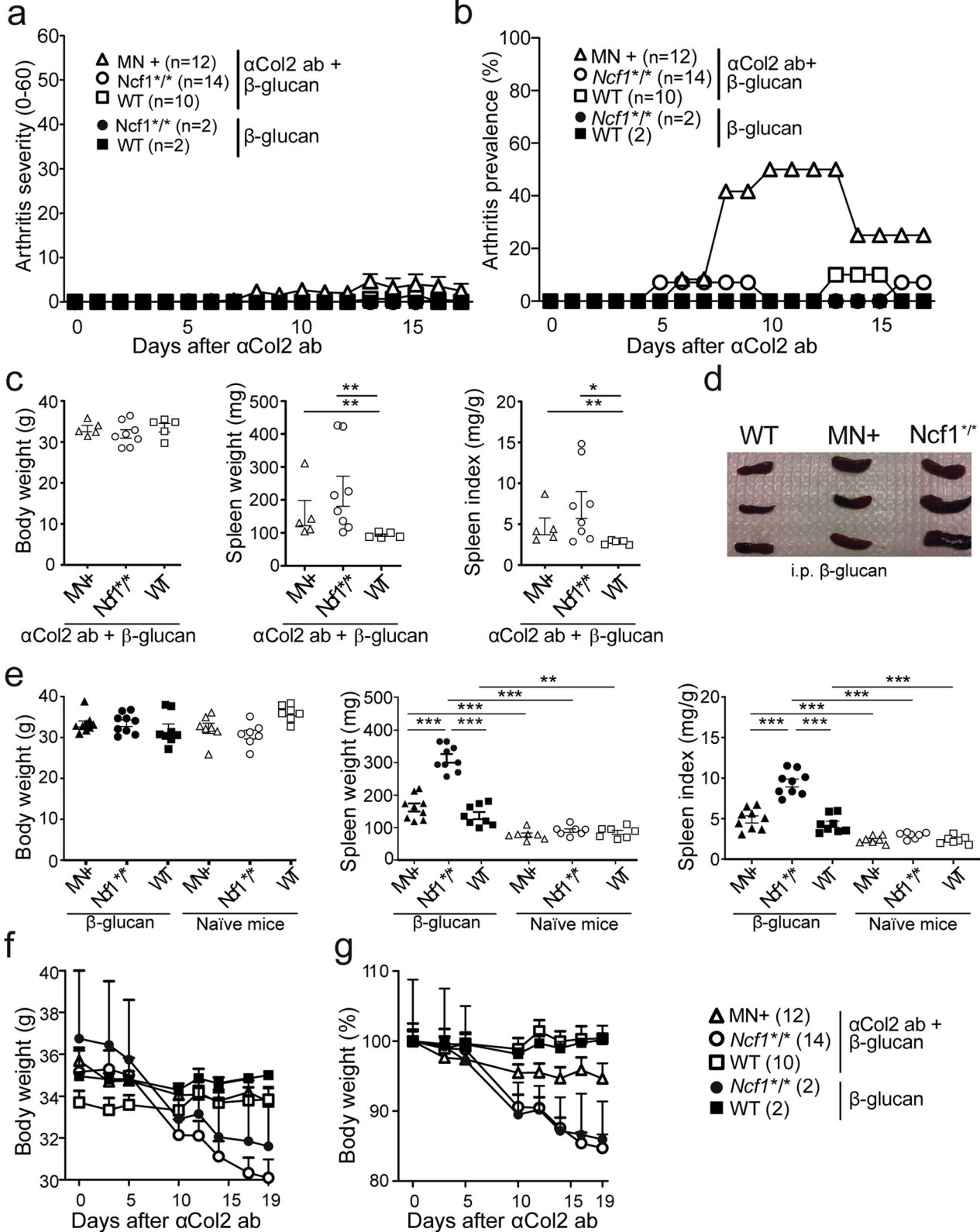

extraction is not capable of inducing arthritis either, even at as high a dose as 2 mg. However, administration of 1,3–1,6-β-glucan induced splenomegaly, which was severely enhanced in mice with a loss of function mutation (m1j) in *Ncf1* (*Ncf1\*/\**), with and without aCol2 antibodies (Fig. 1c–e). Furthermore, 1,3–1,6-β-glucan caused the *Ncf1\*/\** mice, but not wild-type or MN + mice, having functional *Ncf1* in macrophages only, to lose weight

rapidly after injection (Fig. 1f, g). Thus, β-glucan induced severe weight loss if the mice had defective *Ncf1* in macrophages but could not induce PsA or RA-like disease, such as MIP or mCAIA[4,7].

**Pustulan, a 1,6-β-glucan, is protective against mannan-induced PsA-like symptoms.** To further test the ability of β-glucans to

**Fig. 1 Ncf1 deficiency sensitized mice to the effects of β-glucan without enhancing clinical arthritis in CAIA.** Arthritis severity (**a**), prevalence (**b**) was assessed in mice exposed to CAIA initiated by i.v. anti-collagen type II antibodies (aCol2 ab) and triggered by i.p. 1,3–1,6-β-glucan on day 5 to compare Ncf1-deficient (Ncf1*/*) mice and wild-type mice with MN + mice having functional Ncf1 expressed under human CD68 promoter. The mice and the spleens were weighted at euthanasia day after β-glucan in combination with aCol2 ab, and spleen indexes calculated to correlate the spleen weight with the bodyweight of the mouse in CAIA (**c**). Mice and the spleens were weighed also after intraperitoneal β-glucan exposure. Representative images of splenomegaly are shown in (**d**). Bodyweight, spleen weight, and the spleen index are shown in (**e**). The bodyweight curves of mice exposed to CAIA initiated by i.v. anti-collagen type II antibodies (aCol2 ab) and triggered by i.p. 1,3–1,6-β-glucan on day 5 is shown in (**f, g**). Data are pooled from two experiments in (**a, b** and **f, g**); data in (**c** and **e**) are pooled from several experiments. Statistical analyses were performed by Mann–Whitney test. *P < 0.05; **P < 0.01; and ***P < 0.001. Results are presented as mean ± SEM.

regulate PsA we selected to use curdlan, a 1,3-β-glucan from *Alcaligenes faecalis* and pustulan, a 1,6-β-glucan from *Lasallia pustulata*. These compounds were selected as representatives of the 1,3- and 1,6-β-glucans because they are water-insoluble β-glucans but in contrast to glucan extracted from *S. cerevisiae*, these are extracted from bacteria or fungi (lichen). In contrast to branched 1,3–1,6-β-glucan, curdlan and pustulan are linear polysaccharides (Table 1). Curdlan (3 mg) has previously been shown to trigger arthritis in SKG and BALB/c mice[18], and also triggers spondylarthritis in SKG mice[17].

We tested the in vivo effect of 1,3-β-glucan and 1,6-β-glucan in BQ.Ncf1*/* mice. The results showed that neither 1,3-β-glucan (Fig. 2a) nor 1,6-β-glucan (Fig. 2c) could induce arthritis by themselves whereas co-injecting them with mannan down-regulated arthritis symptoms in MIP. The same was seen with the highest dose (3 mg) in arthritis prevalence (Supplementary Fig. 4a, b). The effect seemed to be dose-dependent as there was a trend that 1 and 2 mg of 1,6-β-glucan had less effect on arthritis compared to 3 mg (Fig. 2c and Supplementary Fig. 4b). Both 1,3-β-glucan and 1,6-β-glucan caused splenomegaly at a dose of 3 mg (Fig. 2b, d), while 2 mg 1,6-β-glucan (Fig. 2d) did not. When spleen indexes were calculated to eliminate differences in bodyweight, splenomegaly remained significant with 3 mg of 1,6-β-glucan when administered together with mannan (Fig. 2d). Furthermore, testing the dose-dependency of 1,3-β-glucan with 3 mg and 0.1 mg doses in parallel demonstrated that 0.1 mg dose had no effect on arthritic or psoriatic symptoms in MIP (Supplementary Figs. 5 and 6). None of the β-glucans caused statistically significant weight loss (Supplementary Fig. 7), however, at 3 mg some mice begin to lose some weight. When injection of 1,6-β-glucan (3 mg) was given one day after the mannan injection, there was also a profound regulatory effect with decreased severity (Fig. 2e) and faster healing (Supplementary Fig. 6c) indicating a protective effect of β-glucan also after disease initiation. The mice treated with 1,6-β-glucan 1 day after mannan injection showed an increased splenomegaly (Fig. 2e) consistent with the finding when co-injected with mannan.

**β-glucan (S. cerevisiae) mediates MMR-dependent long-term protection against MIP.** MIP is an acute disease model, however, it has been shown that repeated mannan injections causes new disease flares, similar to the first, after the initial disease has healed[4]. This model also mimics the human disease where the patients have periods of feeling well and periods where the disease flares[34]. We had seen amelioration of MIP in the presence of β-glucans (Fig. 2, above) when disease was induced together/ simultaneously, or one day before, the administration of β-glucan. To investigate whether β-glucan exerts long-term protection we initiated MIP with or without 1,3–1,6-β-glucan with a control group only receiving 1,3–1,6-β-glucan. We thereafter waited until the disease had healed and re-challenged the mice with mannan. The results showed that the β-glucan mediated protective effect was still active after 18 days; the β-glucan pre-treated mice

developed less severe MIP (Fig. 3a) with decreased disease prevalence (Supplementary Fig. 8a).

To investigate which cells mediated the protective effect we investigated macrophages since we had earlier shown that macrophages are key regulatory cells in MIP[4]. Furthermore, we have previously shown that the macrophage mannose receptor (MMR, CD206) mediates a ROS-dependent regulatory effect on MIP[10]. Interestingly, MMR-deficient mice showed a reduced protective effect by 1,3–1,6-β-glucan against MIP, both if injected 18 days prior to or simultaneously with mannan (Fig. 3b and Supplementary Fig. 8b). MMR deficiency affected both the severity and prevalence of the disease. β-glucan caused spleno-megaly in Ncf1-mutated mice, independent of MMR (Supplementary Fig. 8e). No protective effect of 1,6-β-glucan was seen in ROS-sufficient MMR-deficient mice in severity (Fig. 3c) or prevalence (Supplementary Fig. 8c) after re-stimulation with mannan at day 18. Interestingly, treatment with 1,6-β-glucan of mice expressing Ncf1 with a human CD68 promoter (MN + mice), leading to ROS production in macrophages only[35,36], showed that the β-glucan is no longer an effective protector against the severity (Fig. 3d) nor the prevalence of MIP (Supplementary Fig. 8d). This shows that the protective effect is mediated by macrophages. MN + mice did not develop spleno-megaly if subjected to β-glucan (Supplementary Fig. 8f), further confirming the role of macrophages.

**Pustulan, a 1,6-β-glucan, protects against psoriasis.** Pustulan ameliorated arthritic symptoms in the joints in MIP in Ncf1-mutated mice (Fig. 3d), as described above. To investigate whe-ther a 1,6-β-glucan also modulates the psoriatic symptoms in MIP, psoriatic lesions were scored as described in "Methods" in mice exposed simultaneously to mannan and pustulan, the 1,6-β-glucan. Mannan-induced psoriasis was significantly ameliorated by the 1,6-β-glucan in Ncf1-mutated mice (Fig. 4a, b). Similar to PsA (Fig. 3d), the effect was macrophage dependent as the pro-tection was lost in MN + mice (Fig. 4a, b).

The development of psoriatic lesions and arthritis symptoms in MIP is dependent on cytokines such as TNF-α and IL-17 which get expressed a few days after disease induction[4]. To investigate if β-glucan can ameliorate MIP even after the initial triggering phase, 1,6-β-glucan was injected one day after the injection of mannan. The psoriatic lesions were delayed (Fig. 4c, d), indicating a protective role of β-glucan also in already initiated disease. Lack of MMR prohibited 1,6-β-glucans from being protective against psoriasis (Fig. 4e, f). Thus, the effect on psoriasis was similar as observed for arthritis symptoms in MIP.

**Pustulan, a 1,6-β-glucan, downregulates peritoneal resident macrophages.** Peritoneal resident and infiltrating macrophages are among the first defense from an intraperitoneal injection, and it is also part of the first defense when ingesting bacteria con-taining mannan or β-glucans. To further investigate the immu-nomodulatory effects of pustulan, a 1,6-β-glucan, we evaluated its

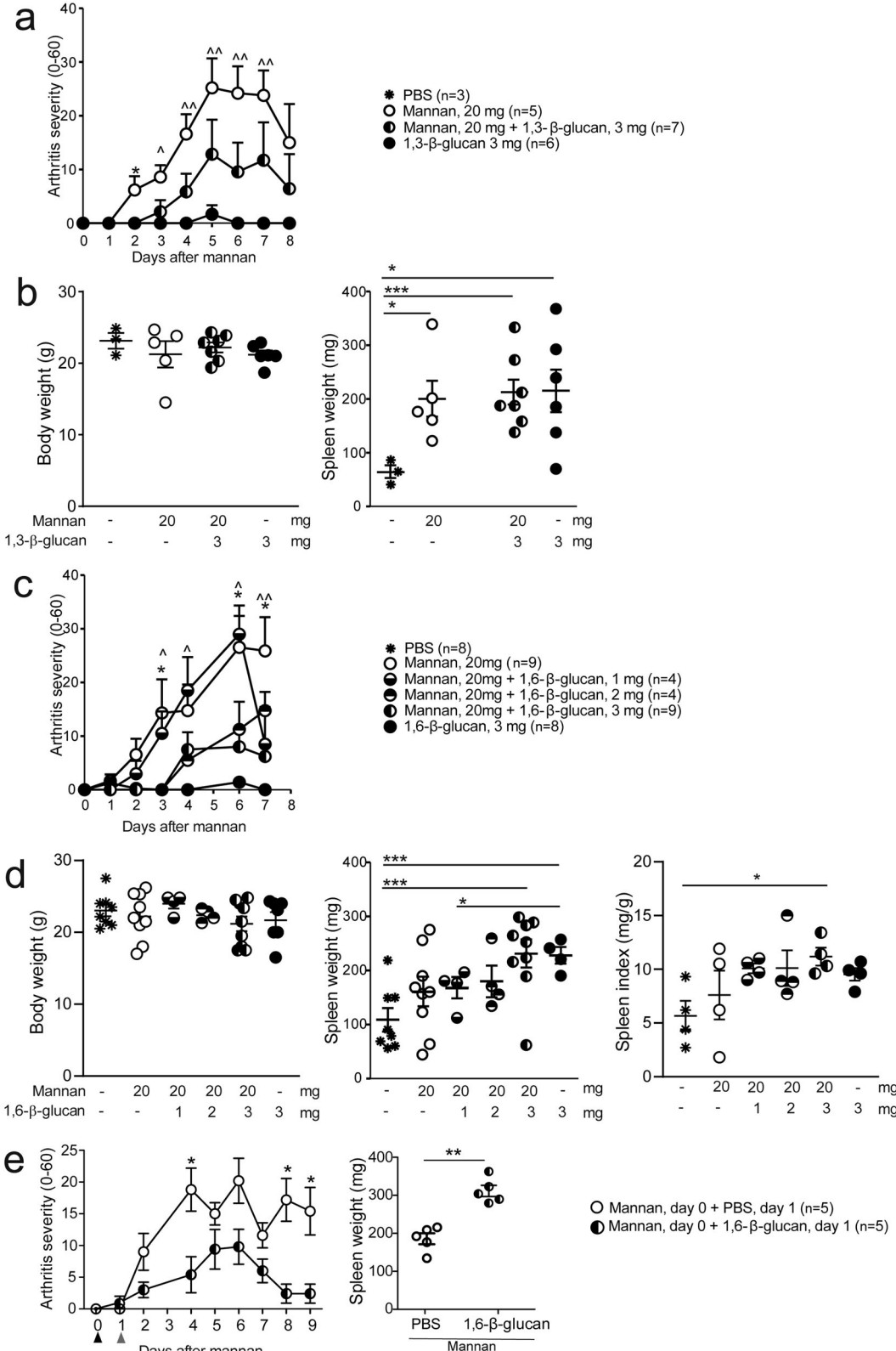

local effects on the cells in the peritoneum where it was administered. To investigate the macrophages phenotypes after exposure to mannan and/or β-glucan, peritoneal resident (CD11b + F4/80hi) and infiltrating (CD11b + F4/80lo) cells were measured at day 8 post injection of either mannan or mannan combined with 1,6-β-glucan. Interestingly, β-glucan lowered the percentages of resident macrophages (Fig. 5a) but increases the

percentages of infiltrating macrophages (Fig. 5b). To investigate if this effect was dependent upon when β-glucan was administrated, peritoneal cells were stimulated in vitro with mannan and/or 1,6-β-glucan for a total of 2 h. Interestingly, 1,6-β-glucan lowers the relative numbers of resident F4/80+ macrophages independent if they were subject to 1,6-β-glucan simultaneously with mannan or 1 h before or after (Fig. 5c). This effect was

**Fig. 2 β-glucan can protect against mannan-induced psoriasis (MIP).** MIP was induced in *Ncf1*-deficient mice with or without co-administration of 1,3-β-glucan (**a**, **b**) or 1,6-β-glucan (**c**, **d**). Arthritis severity was followed, and spleen weight measured at the end of the experiment at day 8 in (**a**, **b**) and day 7 in (**c**, **d**). MIP was induced in *Ncf1*-deficient mice on day 0, and 3 mg of 1,6-β-glucan administered one day later (**e**). Arthritis severity was followed, and spleens weighted at the end of the experiment on day 9. The number of mice in each group is shown in parentheses. The data in (**c**) are pooled from two experiments, except the doses of 1 mg and 2 mg which were included only in one experiment. Spleen index was calculated with the data from one experiment. For groups with 1,6-β-glucan alone, $N = 8$ for arthritis, and $N = 4$ for spleen weight. Results are presented as mean and error bars represent the SEM. Statistical analyses were performed by Mann–Whitney test and * indicates a comparison between mannan and mannan + 3 mg β-glucan, whereas ˆ indicates comparison between mannan and β-glucan in severity graphs. */ˆ$P < 0.05$; **/ˆˆ$P < 0.01$; and ***$P < 0.001$.

independent of whether the mice expressed MMR or not. Contrasting to the in vivo result, 1,6-β-glucan decreased also the relative numbers of CD11b + F4/80lo cells when treated in vitro, independently of MMR (Fig. 5d). The contrasting results are most likely an artifact of the in vitro setting, in which the cellular exchange between tissues is prohibited, in contrast to an in vivo setting.

## Discussion

In this work, we show an anti-inflammatory effect of β-glucans, using the mannan-induced model for psoriasis and psoriatic arthritis in mice. The effect was both preventive (injected 18 days before initiation of disease) and directly interacting with the priming phase (injected simultaneously or one day after mannan injection). The effects were not restricted to certain β-glucan variant, since all three β-glucans tested proved to have downregulative effects. β-glucan activation of macrophages contributed to the protective effect and MMR could be shown to play a role in the beta-glucan mediated protection. None of the β-glucans could induce arthritis nor psoriasis-like symptoms, although some mice developed mild transient swelling and redness of singular joints, but not significantly different from what could occur in normal mice. However, we used C57/Black mice, and it is possible that shifting to other genetic backgrounds could give a more pathogenic effect of β-glucans, e. g. explaining previous results that 1,3-β-glucan could induce transient disease in BALB/c mice[18] and that 1,6-β-glucan could induce arthritis and spondylarthritis in SKG mice[16,17]. Interestingly, β-1,3-glucans have been shown to protect mice from a lethal dose of *Listeria monocytogenes*, if administered prophylactically for 10 days. It was however unclear whether β-glucan reduced the pathogenicity of the pathogen or raised the immune defense[37]. β-glucans has also been studied for their abilities to enhance therapeutic effects of drugs against cancer through uptake by macrophages[38,39], although neutrophils and dendritic cells express receptors for binding β-glucans as well. Macrophages has been shown to be one of the more important immune regulatory cells in MIP, having both the ability to drive[4] the disease and to downregulate it[10]. Considering this, it was not surprising to find that β-glucan can affect the levels of resident macrophages in the peritoneum.

The hyperinflammatory symptoms such as splenomegaly presented in the mice upon β-glucan challenge has previously been indicated both in wild-type mice and in a ROS deficient environment, linked to innate immune cell infiltration and activity of primarily monocytes/macrophages and a production of TNF and IL-2[19,40–42]. Interestingly, many yeast and mold-related diseases are aggravated by defects in the NOX complex[21].

*Ncf1* has previously been shown to have a regulatory role in MIP, where lower NOX2 induced ROS enhances disease[4]. In this model, ROS from macrophages seems to prohibit the β-glucan disease-protective effect. It is thus probable that overreactive macrophages, due to a loss of the downregulative effects of ROS, is downregulated by β-glucan. A connection between ROS produced by NOX2 and MMR was previously shown[10] and in this study, we show that lack of MMR also prohibited the downregulative effect of β-glucans upon MIP. Although MMR has been

shown to regulate MIP[10] the receptor is not currently known to bind β-glucans so it is most likely downstream effects rather than mannan and β-glucan competing over the same receptor. Indeed, injecting mannan on one side of the peritoneum and β-glucan on the other side directly after each other shows the same downregulative effect on MIP (Supplementary Figs. 1–3). It is further confirmed with the long-term protective effects of glucan against MIP and that glucan injected 1-day post disease induction milder the disease. This also indicates that it is not mannan and β-glucan aggregating with each other, and thus cannot induce disease properly, that lowers the severity of MIP. Further confirmed by the protective role of β-glucan even if injected 18 days prior to mannan exposure or one day after.

The purity of the in vivo administered β-glucans had been, to certain extent, characterized by the manufacturers (Table 1). Curdlan and pustulan were tested to induce TLR2 and TLR4 activity when used at concentrations higher than 100 μg/ml and 10 μg/ml, respectively. We thus tested the endotoxin activity in the three β-glucans in parallel using a LAL assay. Glucan from baker's yeast, the 1,3–1,6-β-glucan, was branded 98% pure, and was found free from endotoxins. Endotoxin activity in the 1,3- and 1,6-β-glucan products was found to be 8 EU/mg and 16 EU/mg, respectively. Considering the endotoxin activity of LPS (~25 EU/ng[43]), the detected endotoxin activities would correspond to a LPS contamination of <1 ng/mg, i.e., maximum of 0.0001% impurity. Since the interference of β-glucans with the LAL assay is a well-known phenomenon[28,29], and the b-G-Blocker is indicated in cases where β-glucan contamination is suspected, it is possible that the blocker did not result in complete elimination of the β-glucan's false positive signal in the LAL assay. Taken together, the results showed that the endotoxin activity is so low that the observed in vivo effects of β-glucans could not be due to endotoxin effects. This conclusion is further supported by our observation that β-glucan (*a* 1,3−1,6 the β-glucan from *S. cerevisiae*) has no endotoxin contamination and still do not induce arthritis and mediate protection, the observations and conclusions in the paper is not affected by endotoxin.

Psoriasis, PsA and RA are all complex, polygenic, and heterogenic diseases, suggesting that multiple pathways are involved in the initial stages of the disease and probably also in maintaining the disease. Disease expression involve more than one irregular signal altering innate and adaptive immune regulation and cause autoimmune disease. Here we show that certain β-glucan polysaccharides can protect against mannan-induced psoriasis, rather than inducing disease. Further investigation into how the environment both can induce, but also suppress the development of autoimmune disease, is needed. However, our findings suggest that certain β-glucans have a therapeutic potential in the treatment of autoimmune diseases.

## Methods

**Mice.** BQ.*Ncf1*[m1J] mice (denoted here as *Ncf1*[*/*])[44] and MN + mice with an *Ncf1*[m1J] mutation, but expressing WT *Ncf1* under human CD68 promoter[36] have been described earlier, as has the CD206 (macrophage mannose receptor, MMR) deficient mice[45]. The mice have been back-crossed more than 10 times to BQ background and were used here with (*Ncf1*[*/*].*MMR*[−/−]) or without (*MMR*[−/−]) the

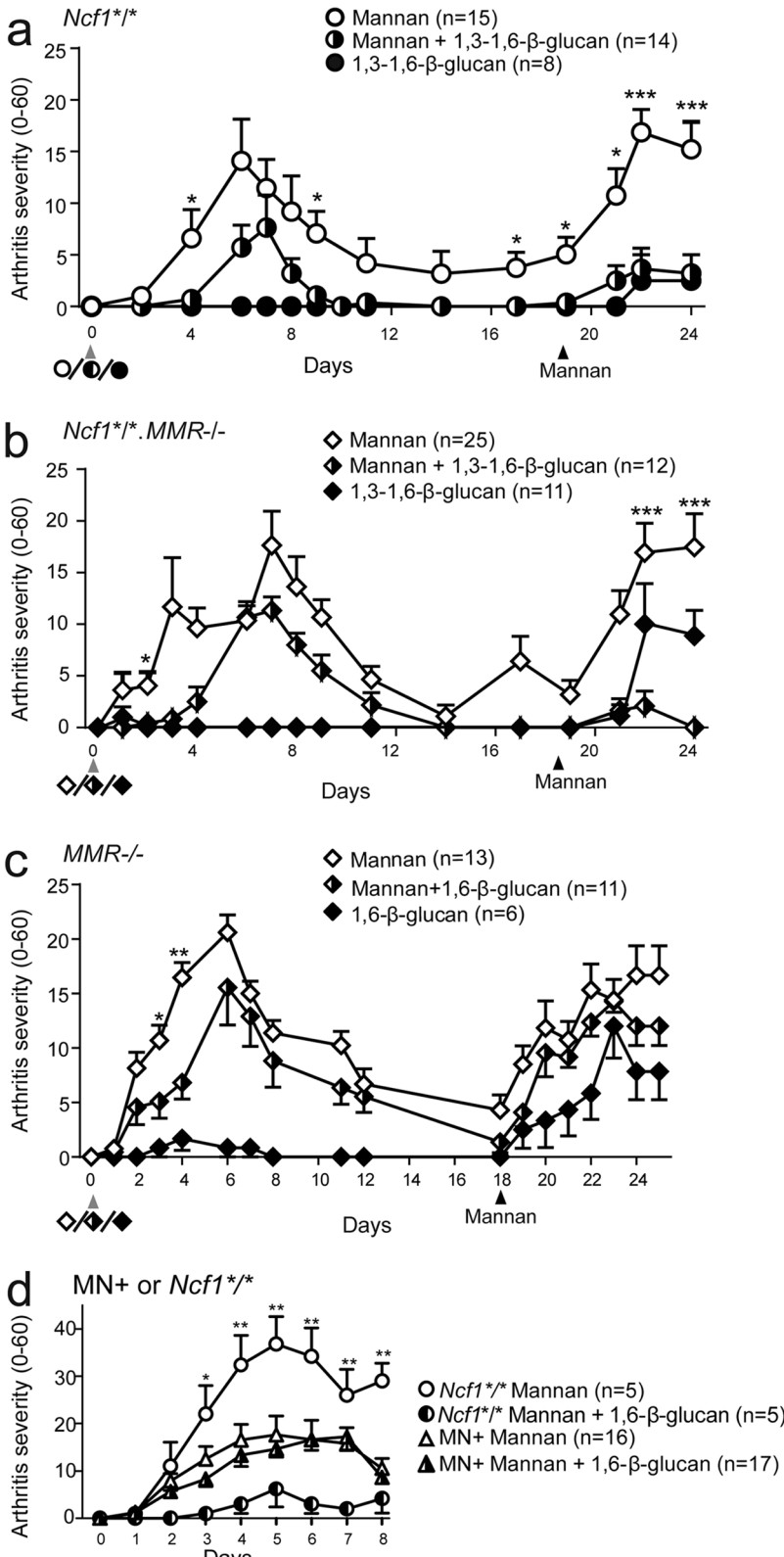

**Fig. 3 Macrophages mediate the protective effect of β-glucan in MIP.** Arthritis severity was followed in MIP induced in *Ncf1*-deficient (**a**), *Ncf1* and *MMR*-deficient (**b**) or *MMR*-deficient (**c**) mice with or without co-administration of 1,3–1,6-β-glucan (5 mg; **a**, **b**) or 1,6-β-glucan (2 mg; **c**). After 18 days, MIP was re-induced by the administration of mannan to investigate the long-term effects of β-glucans. MIP was induced in MN + and *Ncf1*-deficient mice (**d**) with or without co-administration of 1,6-β-glucan (2 mg) to investigate the effect of *Ncf1* expressed under the human CD68 promoter on the *Ncf1*-deficient background in MN + mice. The data are from three combined experiments in (**a**, **b**, **d**), and from one experiment in (**c**, **d**). Number of mice in each group is shown in parentheses in the legend. Statistical analyses were performed by Mann–Whitney test to compare the group receiving mannan with the group receiving mannan and β-glucan. *$P < 0.05$; **$P < 0.01$; and ***$P < 0.001$. Results are presented as mean and error bars represent the SEM.

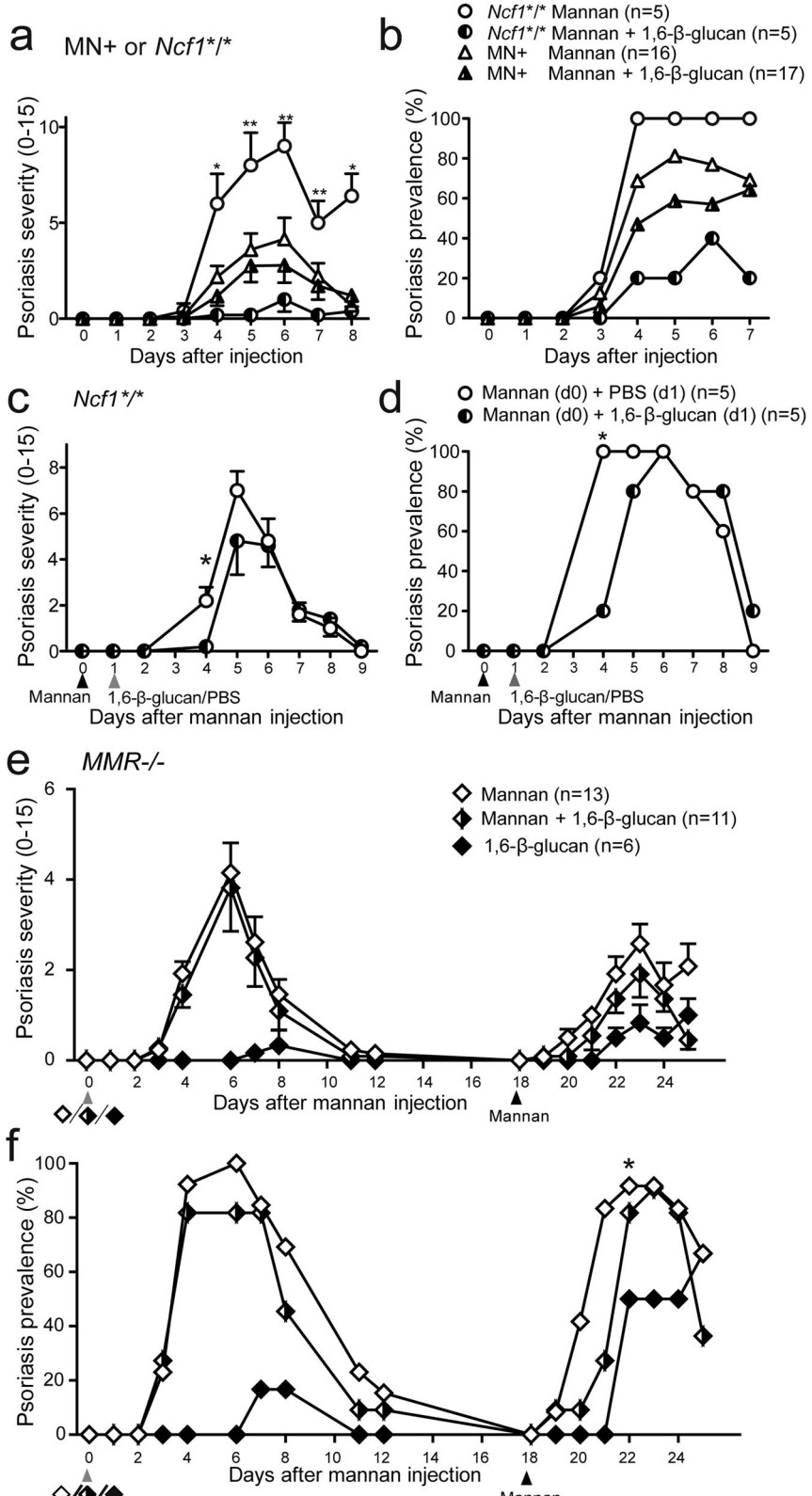

*Ncf1*m1J mutation. B10.Q mice with normal *MMR* and *Ncf1* genes were regarded as WT. The mice were genotyped as previously described[10,36,44,46]. All the mice were housed in specific pathogen-free conditions in open[47] or closed[4] cages, as previously described and provided with enrichments, standard chow, and water ad libitum at the Central Animal Laboratory of University of Turku or at the Medical Inflammation Research animal house at the Karolinska Institute. All mouse experiments followed the ARRIVE guidelines including littermate controls, blind experiments, mixing in cages as well as age- and sex balance between groups.

Exclusion was only done if the mice were suffering severe weight loss or exhibited other signs of poor well-being. Experiments were performed with 8–25 weeks old mice. Only naïve mice were used, no previous experiments have been performed on them.

**Polysaccharides**. The polysaccharides used in the study are described in Table 1. Mannan (#M7504) and 1,3–1,6-β-glucan (#G5011) from *Saccharomyces cerevisiae*

**Fig. 4 Psoriatic symptoms are ameliorated by 1,6-β-glucan.** MIP was induced in *Ncf1*-deficient mice and in MN + mice (*Ncf1*\*/\* mice expressing functional *Ncf1* under a human CD68 promotor) with or without co-administration of 1,6-β-glucan (2 mg) and the severity (**a**) and the prevalence (**b**) of psoriatic lesions were quantified by macroscopic scoring. The severity (**c**) and prevalence (**d**) of psoriatic lesions was followed in *Ncf1*-deficient mice when 1,6-β-glucan (2 mg) was administered one day after induction of MIP. Development of psoriatic lesions was investigated also in MMR-deficient mice when MIP was induced with or without co-administration of 1,6-β-glucan (2 mg) and when MIP was re-induced by mannan 18 days later (**e**, **f**). Number of mice in each group are shown in parentheses. Data are pooled from two (**a**–**d**) or one (**e**, **f**) experiments. Statistical analyses were performed by Mann–Whitney test to compare the group receiving mannan with the group receiving mannan and β-glucan. \*P < 0.05; \*\*P < 0.01. Values are presented as mean and error bars present the SEM.

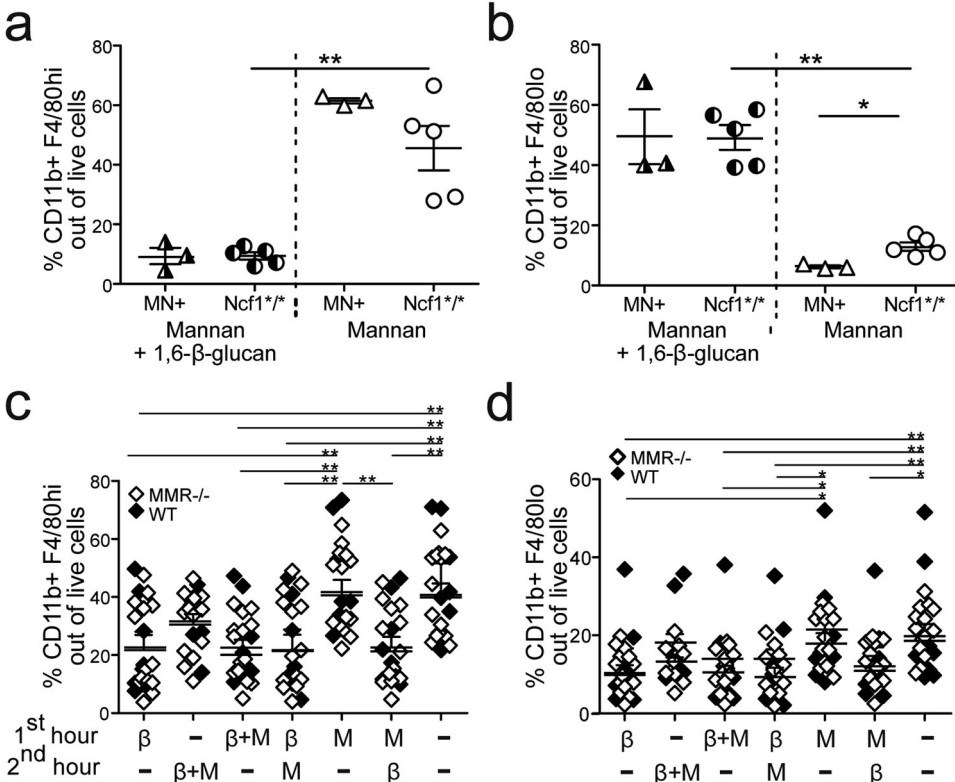

**Fig. 5 1,6-β-glucan decreases the number of resident peritoneal macrophages.** Resident (**a**) and infiltrating (**b**) macrophages were measured among live peritoneal cells collected day 8 after induction of MIP with or without co-administration of 1,6-β-glucan (2 mg) in *Ncf1*\*/\* and MN + mice. For analysis of cell phenotypes after in vitro stimulation (**c**, **d**), naïve peritoneal cells were collected from MMR$^{-/-}$ and WT mice and challenged in vitro for a total of 2 h, using two different stimulation points (1 h in between), with mannan (M) and/or 1,6-β-glucan (β) without washes between the treatments. Resident macrophages were defined as CD11b + F4/80hi (in **a** and **c**) and infiltrating macrophages as CD11b + F4/80lo (in **b** and **d**), as determined by flow cytometry. Data in (**c**, **d**) are a combination of two experiments with total n = 21. Statistical analyses were performed with Mann–Whitney test (**a**, **b**) and one-way ANOVA using Bonferroni post test (**c**, **d**). \*P < 0.05; and \*\*P < 0.01. Results are presented as mean with SEM.

were purchased from Sigma-Aldrich. 1,3-β-glucan from *Alcaligenes faecalis* (Curdlan, #tlrl-curd) and 1,6-β-glucan from *Lasallia pustulata* (Pustulan, #tlrl-pst) were purchased from InvivoGen.

Limulus Amebocyte Lysate (LAL) assayLimulus Amebocyte Lysate (LAL) assay (QCL-1000®, #50-647U by Lonza, Walkersville, USA) was used to determine the endotoxin concentration in aqueous suspensions of β-glucans, in the presence or in the absence of β-G-Blocker (B50-700, Lonza, Walkersville, USA). Serial dilutions were tested up to 5 mg/ml (1,3–1,6-β-glucan) or 3 mg/ml (1,3- and 1,6-β-glucans). The absorbance of the chromogenic substrate was read at 405 nm with a Synergy 2 Multi-Mode Microplate Reader with BioTek Gen5 software (BioTek Instruments, Winooski, VT, USA). Endotoxin concentration was determined with the help of the standard and using the β-glucan dilutions that resulted in an absorbance result within the standard curve.

**In vivo administrations.** For induction of CAIA, a cocktail of four monoclonal anti-collagen type II (aCol2) antibodies, produced in-house from the clones U1, M2139, CIIC1 and CIIC2, were injected intravenously according to a previously reported protocol[33] at day 0. At day 5 post immunization 800 μg of 1,3–1,6-β-glucan, suspended in PBS, was administrated intraperitoneally.

MIP was induced by injecting 10 mg of mannan if open cages and 20 mg mannan if closed cages due to differences in sensitivity toward the compound, as previously described[4,10].

1,3-β-glucan, 1,6-β-glucan or 1,3–1,6-β-glucan were tested alone or in combination with MIP, 18 days before disease induction, simultaneously with induction of MIP or 1-day post MIP induction in different concentrations in PBS suspensions by intraperitoneal injection. All suspensions were well mixed during the administration procedure to ensure each mouse received the same dose.

A small trial was also performed were mice were injected with glucan and mannan at different sides of the peritoneum to rule out that the effect is due to aggregation of the two in the tube prior to injection. Furthermore, a small trial was performed were the aggregates in the glucan suspension was filtered away prior to injection using a 70-μm filter.

**Macroscopic scoring of disease development.** To assess disease, the mice were blindly scored according to a standardized macroscopic scoring system evaluating both arthritic symptoms and psoriatic lesions[4,47,48]. For arthritic symptoms, one point was assigned for each swollen and red toe or knuckle, and five points were assigned for each inflamed ankle, giving maximum of 15 points/paw and 60 points /mouse. For psoriasis, the maximum was 15 points/ mouse. Each mouse was also weighed during the experiment to assess potential weight loss of the mice and the spleens were weighed post euthanasia to assess splenomegaly.

**In vitro treatment.** Naïve peritoneal cells were harvested with PBS, collected by centrifugation, counted, resuspended into DMEM, and equal number of cells were plated to a 96-well plate. The cells were subsequently stimulated with mannan

(5 mg/ml) and/or 1,6-β-glucan (1 mg/ml) or left in plain culture medium without stimuli. After 1 h of incubation at +37 °C, the cells received another round of stimuli by mannan and/or 1,6-β-glucan or medium as control, without being washed in between, resulting in different combinations of stimuli. After another 1-h incubation at +37 °C, the cells were collected for flow cytometry analyses.

**Flow cytometry**. Flow cytometric analyses of peritoneal samples were performed as described earlier[47,49]. After cells had been counted and equally plated, FcγIII (CD16) and FcγII (CD32) receptors were blocked (ratIgG2b blocking of CD16/CD32, #553142, BD Biosciences) and surface antigens were stained with fluorescently labeled antibodies. The antibodies used were CD11b-APC-Cy7 (BD Biosciences #561039) and F4/80-Texas Red-PE (Invitrogen #MF48017) or F4/80-AF647 (Serotec #MCA497A647). The samples were acquired on a Fortessa flow cytometer (BD Biosciences) and the results analyzed using FlowJO software V10 (Tree Star). Gating strategy is illustrated in Supplementary Fig. 9. As a reference anti-CD16/CD32 was diluted 1:80 prior to the addition of 15 μl to cell pellet, after 5 min the other antibodies are diluted 1:100 and 15 μl added to each well and incubated for 20 min on ice (no wash in between).

**Statistics and reproducibility**. Statistical analyses were performed with GraphPad Prism, version 5. Disease scores, weight of mice and spleen data and cellular analyses were analyzed by Mann–Whitney test, and comparisons of multiple groups after in vitro stimulation was performed by one-way ANOVA using Bonferroni post test. Animal numbers are indicated in the figures. Experiments are reproduced 2–3 times. Replicates are defined using identical genotypes and treatment schemes.

**Study approval**. All experiments were carried out under ethical permit numbers ESAVI-0000497/041003/2011 and ESAVI/439/04.10.07/2017 at the University of Turku and the Karolinska Institute under the permit number N490/12 and N35/16.

**Reporting summary**. Further information on research design is available in the Nature Research Reporting Summary linked to this article.

## Data availability

The data that support the findings of this study are available from the corresponding authors upon reasonable request and at https://doi.org/10.17605/OSF.IO/KGYS3, from the www.osf.oi server, accession ID kgys3.

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

## Acknowledgements

Flow cytometry was performed at the Cell Imaging Core, Turku Centre for Biotechnology, Turku, Finland. We would like to thank Academy Professor Sirpa Jalkanen for scientific discussions and Riina Larmo for excellent animal caretaking. We also thank our funding agenecies: The Academy of Finland, the Sigrid Jusélius Foundation, the National Doctoral Programme in Informational and Structural Biology, the Drug Research Doctoral Programme of University of Turku Graduate School, the State Research Funding of Turku University Hospital, the Turku University Foundation, the King Gustaf V 80 Years Foundation, the Finnish Cultural Foundation, the Varsinais-Suomi Regional Fund, the Swedish Foundation for Strategic Research, the KA Wallenberg Foundation, and the Swedish Research Council. The funding sources were not involved in the study design; in the collection, analysis, or interpretation of data; in the writing of this article; or in the decision to submit the article for publication.

## Author contributions

C.F.-H.: designing research studies, performing experiments, acquiring of data, data analysis, writing the manuscript. O.S.: designing research studies, performing experiments, acquiring data, data analysis, revising the manuscript. S.R.; performing experiments, acquiring of data, data analysis. R.H.: supervision and overall responsibility, designing research studies, writing the manuscript. All authors ensure the accuracy of this work.

## Funding

## Competing interests

The authors declare no competing interests.
