## [Peer Review File · Communications Biology]

Reviewers' comments:

Reviewer #1 (Remarks to the Author):

The manuscript from Hagert and collaborators intends to evaluate the effect of different beta-glucans on the outcome of mannan-induced psoriasis like symptoms in mice. To do that, the authors follow a number of biological responses, including the disease progression, arthritis severity, recruitment of macrophages in WT and Ncf1 deficient mice after injection of large quantities of commercially available mannans and beta-glucans from diverse origins. The study involves a considerable amount of work to achieve statistical significance in different background. Although the role of glucans in therapeutic usage is of potentially great interest, the study suffers from a number of methodological shortcomings that need to be resolved to validate the results.

- the beta-glucans are notoriously insoluble polysaccharides that are very difficult to use in solution and that are for this purpose often depolymerized or solubilized by chemical modification in order to be used. When in aqueous solution, the form clumps and a part of the glucans are in suspension rather than truly solubilized. It is well known that the soluble and the insoluble fractions of beta-glucans exert different biological activities, that may be antagonist. Thus, before injecting these compounds their solubility should be assayed.

- the study uses commercially available products which structure and purity have not been assayed and that have not been prepared for internal usage. As such, they may contain a number of impurities, and worse bioactive bacterial/fungal derived products that may interfere with results. Yeast beta glucan is branded as 98% pure, while no indication is given for the others. According to retailer, the presence of bacterial contamination was only assessed for 1,6betaGlucan. Finally, no batch control is mentioned in the methods section. It appears thus impossible to use such product in vivo in animals without a throughout quality check and structural analysis.

- The authors fail to explain the rationale of using one or another beta-glucan throughout the study. The manuscript seems to convey the idea that 1,3-1,6betaGlucan is the combination of 1,3betaGlucan and 1,6betaGlucan, which is not true as the three polysaccharides show very different structures, sizes and conformations. As a result, although the author use three different molecules, no attempt is made to obtain structure-to-function relationship information.

- The authors insufficiently evaluate the biological in vivo relevance of injecting large quantity of mannan (10mg) and glucan (0.8mg) in comparison with serum concentration of polysaccharides in case of bacterial or fungal infection responsible for PsA triggering.

- Fig 3B contains several potential problems. First the labels are different from other Figures which make the data not consistent. Second, there seems to be a curve switch between betaGlucan and Mannan+betaGlucan at day 6, but because the two points are superimposed, it is not possible to evaluate and should be checked.

Reviewer #2 (Remarks to the Author):

The manuscript by Hagert et al. reports on the effects of different types of β -glucan on the regulation of autoimmune inflammation in mice with CAIA or MIP. This is an important area, and there is much new and useful information here, but substantial revision is needed.

1. The authors declared the administration of 1,3-1,6- β -glucan induced splenomegaly in Ncf1^{*/*} mice, with and without aCol2 antibodies with Figure 1C-D. Please provide the body weight data at euthanasia day, and calculate the spleen index (spleen weight/body weight), to be more accurate. Figure 2A-B as well. Besides, please justify why chose the 1,3-1,6- β -glucan to address the possibility that β -glucans could induce arthritis in CAIA, while tested the other two variants in MIP.

2. Regarding the dosage, why immunized with 800 μ g 1,3-1,6- β -glucan in Fig 1, and administered with 3 mg of 1,3- β -glucan or 1,6- β -glucan? Is it reasonable to test 20 mg β -glucan in MIP referred to 20 mg Mannan? Otherwise, can you provide the molecular weight of Mannan and three types of β -glucan?

3. Clarification is needed to understand the conclusion that 'the anti-inflammatory effect of β -glucans, are likely not dependent on the beta glucan polymetric variants, since all three β -glucans tested proved to have down-regulative effects in a dose-dependent manner.' However, I can only see the results of 1, 2, 3 mg of 1,6- β -glucan to potentially demonstrate a dose-dependent manner.

In summary, I think it is a nice study with some original, interesting data on the protective effect of β -glucans in the context of mannan-induced psoriasis and psoriatic arthritis.

Specific comments:

1. Line 115 (Figure 1 and 2A and Suppl. Fig 1A)
2. Line 64-65 please add the references, or highlight it is in Ncf1^{*/*} mice?
3. In Fig 3 and 4, the symbols for significant difference make me feel crazy, honestly!
4. In the abstract, " β -glucans cannot induce arthritis, psoriasis or PsA in wild-type mice.", which can't be fully supported by current data in the light of dosage. Indeed, I prefer to see some information about the mice used in this study in the abstract.

Reviewer #1 (Remarks to the Author):

- 1) *the beta-glucans are notoriously insoluble polysaccharides that are very difficult to use in solution and that are for this purpose often depolymerized or solubilized by chemical modification in order to be used. When in aqueous solution, the form clumps and a part of the glucans are in suspension rather than truly solubilized. It is well known that the soluble and the insoluble fractions of beta-glucans exert different biological activities, that may be antagonist. Thus, before injecting these compounds their solubility should be assayed.*

AUTHOR RESPONSE:

As pointed out the solubility of the used polyglycans varies and we have now described this in more detail. According to the manufacturer, glucan from baker's yeast (G5011, see Table 1) is soluble in water (9.80 - 10.20 mg/ml). We used this glucan dissolved in PBS at concentration ≤ 5 mg/ml. In contrast, the 1,3-beta-glucan (tlr-curd) and 1,6-beta-glucan (tlr-pst) are water-insoluble (See the new information in Table 1). However, they can both be prepared to form a suspension which results in a suspension with gelatinous precipitates, per the manufacturer's technical data sheet. In this study, these compounds were administered as a newly mixed suspension at a concentration of ≤ 3 mg/ml. We have now corrected the wording throughout the manuscript to describe more accurately whether a solution or a suspension of the compounds were used, including terms such as "solubilized", "dissolved" or "suspended".

To test the biological activities of the soluble and the insoluble fractions of beta-glucans we tested the 3mg/ml suspension of 1,3-beta-glucan (tlr-curd) in parallel with a filtered (40 μ m) suspension. The results point to that the undissolved particles are important for the biological effect of 1,3-beta-glucan. The filtered suspension seems to have a milder effect on mannan-induced arthritis (Figure below), irrespective of whether it was administered in different (A-B) the same (C-D) site with mannan. Milder effect with filtered beta-glucan was also observed in psoriasis symptoms and body weight. These results reflect the dose-dependent effect of beta-glucans (Figure 2) and highlight the importance of administering a well-mixed suspension to ensure equal dosing. A comment on that has been added into the methods section in paragraph "In vivo administrations".

Figure 1. Comparison of filtered and unfiltered glucan. A-B) A suspension of curdlan (1,3- β -glucan) used unfiltered (oranges squares) and as filtered (brown circle) cannot induce arthritis. In parallel mice were treated with mannan alone (orange diamond) or in combination with filtered (blue half-filled triangles) or unfiltered curdlan (purple half-filled squares). Mannan and curdlan was injected at different sides of the peritoneum. **C-D)** As a comparison (in parallel with A-B), non-filtered (light purple half-filled square) and filtered curdlan (green half-filled circle) in combination with mannan was mixed in the same tube and inject i. p. in simultaneously. Mannan only and unfiltered or filtered curdlan was included as positive, respectively negative control in both settings. The graphs are based on one experiment and includes mice that have been housed in closed cages. Statistics were done with Mann-Whitney T test. Mean \pm SEM. *P<0.05, **P<0.01, ***P<0.001 and were the color indicates which mannan-curdlan combination it is significant against.

Figure 2. Comparison of filtered and unfiltered glucan upon psoriatic lesions. A-B) A suspension of curdlan (1,3- β -glucan) used unfiltered (orange squares) and as filtered (brown circles) cannot induce psoriasis. In parallel mice were treated with mannan alone (orange diamond) or in combination with filtered (blue half-filled triangles) or unfiltered curdlan (purple half-filled squares). Mannan and curdlan was injected at different sides of the peritoneum. **C-D)** As a comparison (in parallel with A-B), non-filtered (light purple half-filled square) and filtered curdlan (green half-filled circle) in combination with mannan as a control was inject i. p. in the mice simultaneously. Mannan only and unfiltered or filtered curdlan was included as positive, respectively negative control in both settings. The graphs are based on one experiment and includes mice that have been housed in closed cages. Statistics were done with Mann-Whitney U test, where mean is \pm SEM. * $P < 0.05$, ** $P < 0.01$ and were the color indicates which mannan-curdlan combination it is significant against.

2) the study uses commercially available products which structure and purity have not been assayed and that have not been prepared for internal usage. As such, they may contain a number of impurities, and worse bioactive bacterial/fungal derived products that may interfere with results. Yeast beta glucan is branded as 98% pure, while no indication is given for the others. According to retailer, the presence of bacterial containment was only assessed for 1,6betaglucan. Finally, no batch control is mentioned in the methods section. It appears thus impossible to use such product in vivo in animals without a throughout quality check and structural analysis.

AUTHOR RESPONSE:

We have now included a table (Table 1) to summarize the used compounds and information on their purity. The purity of the compounds is now also described in the manuscript (rows 287-303).

According to the manufacturer, the presence of bacterial contamination (e.g. lipoproteins and endotoxins) in curdlan and pustulan has been assessed using HEK-Blue™ TLR2 and HEK-Blue™ TLR4 cells. Upon request for additional information on the purity of both curdlan and pustulan, InVivoGen customer support replied that “Regarding the presence of contaminants, we tested the non-activation of TLR-2 or TLR-4 (at 10µg/mL) by using our HEK-Blue TLR cell lines.” Regarding curdlan they wrote “I confirm that even 100µg/mL of Curdlan will not activate TLR2 or TLR4. And regarding pustulan, “Our last batch activate TLR2 at 100µg/mL and TLR4 at 30µg/mL.”

To further characterize the used β-glucans, we performed a Limulus Amebocyte Lysate (LAL) assay to determine possible endotoxin contamination of the products. LAL assay has also been used by others (e.g. Noss et al, Innate Immunity 19(1) 10–19) to test glucans. LAL assay is a sensitive method, and since b-glucans are known to interfere with the assay, we used the commercial b-G-Blocker to block the G pathway in the LAL and to render the assay more specific for endotoxin. Results are now presented in the Results section and as Supplemental Figure 3, and the purity of the products discussed in the Discussion section. Table 1 contains also additional information on the compounds, including the structures of the used beta-glucans.

Manufacturers’ specifications for purity and test results e.g. regarding possible bacterial contamination in the used compounds were considered adequate in relation to study objectives and different batches were not tested in parallel *in vivo*. If unexpected differences in biological effects would have been seen after switching to another lot, troubleshooting would have been initiated to find the root cause.

- The authors fail to explain the rationale of using one or another beta-glucan throughout the study. The manuscript seems to convey the idea that 1,3-1,6betaGlucan is the combination of 1,3betaGlucan and 1,6betaGlucan, which is not true as the three polysaccharides show very different structures, sizes and conformations. As a result, although the author use three different molecules, not attempt is made to obtain structure-to-function relationship information.

AUTHOR RESPONSE:

We have now explained the rationales of using one or another b-glucan in the experiments throughout the Results-section. We also provide below a table to summarize the experiments presented in the paper.

Glucan	Arthritis	Symptoms		
		Psoriasis	Splenomegaly	Weight loss
1,3–1,6-β - glucan (Glucan from baker's yeast)	Ncf1 ^{*/*} MN WT (Fig. 1)		Ncf1 ^{*/*} MN WT (Fig. 1)	Ncf1 ^{*/*} MN WT (Fig. 1)
	Ncf1 ^{*/*} Ncf1 ^{*/*} .MMR ^{-/-} (Fig. 3)		Ncf1 ^{*/*} .MMR ^{-/-} Ncf1 ^{*/*} .MMR ^{+/+} (Suppl. Fig 4)	
1,3-β -glucan (Curdlan)	Ncf1 ^{*/*} (Fig. 2)		Ncf1 ^{*/*} (Fig. 2)	Ncf1 ^{*/*} (Fig. 2)
1,6-β -glucan (Pustulan)	Ncf1 ^{*/*} (Fig. 2)		Ncf1 ^{*/*} (Fig. 2)	Ncf1 ^{*/*} (Fig. 2)
	MMR ^{-/-} (Fig.3)	MMR ^{-/-} (Fig.4)		
	Ncf1 ^{*/*} MN ⁺ (Fig.3)	Ncf1 ^{*/*} MN ⁺ (Fig. 4)		

To avoid conveying the impression that 1,3-1,6-b-Glucan is the combination of 1,3-b-Glucan and 1,6-b-Glucan we have made several clarifications throughout the manuscript, e.g. edited the subtitles in the Results-section.

We tested different beta-glucans to try and assess if there was a difference between them in efficiency, which there is not. Since there is no significant difference in efficiency in treatment of psoriasis nor arthritic symptoms between the different polyglycans, we see no reason to go further in trying to compare them in a structure-to-function manner. We have further chosen not to perform all the experiments with all the different variants of b-glucans for the same reason, since there is no real difference between their in vivo effects in our models. We cannot, with the 3R rule in mind, defend doing the same experiments with all 3 different compounds. A table to summarize the structures of beta-glucans has now been added (See Table 1).

- The authors insufficiently evaluate the biological in vivo relevance of injecting large quantity of mannan (10mg) and glucan (0.8mg) in comparison with serum concentration of polysaccharides in case of bacterial or fungal infection responsible for PsA triggering.

AUTHOR RESPONSE:

We have now discussed the dosing relevance. It is obvious that the dosing is far more than is obtained in exposures from infections or inhalation. However, in the real situation it might be a situation where other environmental factors as well as an unfortunate polygenic situation that could make the individual susceptible to much lower doses. We have only studied an experimental situation and it will be possible to further address interacting environmental and genetic factors also in the experimental situation that could titrate the efficiency of both the mannan induction of disease and the beta-glucan mediated protection in a subsequent study.

- Fig 3B contains several potential problems. First the labels are different from other Figures which make the data not consistent. Second, there seems to be a curve switch between betaGlucan and Mannan+betaGlucan at day 6, but because the two points are superimposed, it is not possible to evaluate and should be checked.

AUTHOR RESPONSE:

The labels are different in shape to highlight the difference in strain background.

The authors thank the reviewer for being so observant. The labels had accidentally been switched around. We are sorry for this and have corrected it. Open diamond should be mannan and filled should be b-glucan.

The original data was revisited to double-check the possible curve switch and there isn't one. Maybe the confusion was a result from the unfortunate mix up in the labels that have been corrected.

Reviewer #2 (Remarks to the Author):

1. *The authors declared the administration of 1,3-1,6- β -glucan induced splenomegaly in *Ncf1*^{*/*} mice, with and without *aCol2* antibodies with Figure 1C-D. Please **provide the body weight data at euthanasia day**, and calculated **the spleen index** (spleen weight/body weight), to be more accurate. Figure 2A-B as well. Besides, please justify why chose the 1,3-1,6- β -glucan to address the possibility that β -glucans could induce arthritis in CAIA, while tested the other two variants in MIP.*

AUTHOR RESPONSE:

We have now provided the body weight data at euthanasia day and calculated the spleen index for **Figure 1C-D** and updated the figure with this information (new Figure 1C-E).

We have now also provided the body weight data at euthanasia day for **Figure 2A-B** and updated the figure with this information (new Figure 2B-D).

For **figure 2B, former 2A** we were not able to locate the traceability link, the cage/ear mark e.g., between spleen weight and body weight and thus spleen index is not calculated.

For figure 2D, former 2B, we could correlate the spleen weights with the body weight in one of the pooled experiments (with N 4-5) and provide it as representative data. The rest of the spleen weight data has traceability to the treatment group they belong to, but not to individual mice and their weights, and thus these are not included.

We have now explained in the manuscript the rationale of choosing the 1,3-1,6- β -glucan for the CAIA-experiment.

2. *Regard with the dosage, why immunized with 800 μ g 1,3-1,6- β -glucan in Fig 1, and administered with 3 mg of 1,3- β -glucan or 1,6- β -glucan? Is it reasonable to test 20 mg β -glucan in MIP referred to 20 mg Mannan? Otherwise, can you provide the molecular weight of Mannan and three types of β -glucan?*

AUTHOR RESPONSE:

We have used 800 μ g 1,3-1,6- β -glucan in Fig 1, and administered 3 mg of 1,3- β -glucan or 1,6- β -glucan because in CAIA, the immunization is done with antibodies, and mice often get symptoms without any further stimuli, which mimics the effector phase of RA with autoantibodies in the serum. Relatively small doses of additional immunostimuli, often in ng or μ g (please see Kelkka et al. for several examples DOI 10.1016/j.ajpath.2012.03.031), increase the severity. As an additional immunostimuli in CAIA we chose to test 800 μ g dose, which is in the range of commonly used doses (ng – μ g), which we now also describe in the manuscript (rows 142-143).

In MIP, we titrated the dose up to 3mg and observed disease amelioration, and thus did not need to increase the dose to answer the study objectives; a decision supported also by the trend in mice losing some weight (suppl Fig 5) with that dose.

This information has now been added also into the manuscript, for clarity (row 173-181). Unfortunately, it is not possible to inject 20mg of B-glucan because of its insolubility in water or in this case PBS. However, b-glucan can be made into a solution with PBS at lower concentration which is what has been done here. The molecular weights of the different compounds have been added to the newly added table 1.

3. *Clarification is needed to understand the conclusion that ‘the anti-inflammatory effect of β -glucans, are likely not dependent on the beta glucan polymeric variants, since all three β -glucans tested proved to have down-regulative effects in a dose-dependent manner.’ However, I can only see the results of 1, 2, 3 mg of 1,6- β -glucan to potentially demonstrate a dose-dependent manner.*

AUTHOR RESPONSE:

We have tested, in parallel, 3 mg and 0,1 mg doses of 1,3-beta-glucan on MIP and seen a significant amelioration with 3 mg but only a trend with 0,1 mg dose (Figures below), when arthritic symptoms (upper panel, A-D) and psoriatic symptoms (lower panel, A-D) were investigated. This information has now been added into the manuscript to further support our claim on dose-dependency, and the claims edited to fit the data.

Figure 3. 0.1mg of B-glucan is insufficient to protect against arthritic disease A-B) A suspension of 3mg curdlan (1,3- β -glucan) injected either at the same side of the peritoneum (open squares) or different sides of the peritoneum (purple squares) compared to 20mg mannan. As a positive respectively negative control, mice were treated with mannan alone (orange) or glucan alone (brown squares). C-D) In a parallel experiment a dose of 0.1mg curdlan was used in combination with mannan, either on same side of peritoneum (purple triangle) or different sides of the peritoneum (blue). As a positive respectively negative control, mice were treated with mannan alone (orange) or glucan alone (green). The graphs are based on one experiment and includes mice that have been housed in closed cages.

Statistics were done with Mann-Whitney U test, where mean is \pm SEM. * $P < 0.05$, ** $P < 0.01$ and were the color indicates which mannan-curdlan combination it is significant against.

Figure 3. 0.1mg of B-glucan is insufficient to protect against psoriatic lesions A-B) A suspension of 3mg curdlan (1,3- β -glucan) injected either at the same side of the peritoneum (open squares) or different sides of the peritoneum (purple squares) compared to 20mg mannan. As a positive respectively negative control, mice were treated with mannan alone (orange) or glucan alone (brown squares). C-D) In a parallel experiment a dose of 0.1mg curdlan was used in combination with mannan, either on same side of peritoneum (purple triangle) or different sides of the peritoneum (blue). As a positive respectively negative control, mice were treated with mannan alone (orange) or glucan alone (green). The graphs are based on one experiment and includes mice that have been housed in closed cages. Statistics were done with Mann-Whitney U test, where mean is \pm SEM. * $P < 0.05$, ** $P < 0.01$ and were the color indicates which mannan-curdlan combination it is significant against.

Specific comments:

1. *Line 115 (Figure 1 and 2A and Suppl. Fig 1A)*

AUTHOR RESPONSE: Thank you for noticing the incorrect figure reference. It has been removed.

2. *Line 64-65 please add the references, or highlight it is in Ncf1^{*/*} mice?*

AUTHOR RESPONSE: The references have been added.

3. *In Fig 3 and 4, the symbols for significant difference make me feel crazy, honestly!*

AUTHOR RESPONSE: We have now edited figures 3 and 4 and kept only the symbols for the significance for the most important comparison, namely that between mannan and mannan + beta-glucan.

4. *In the abstract, “ β -glucans cannot induce arthritis, psoriasis or PsA in wild-type mice.”, which can't be fully supported by current data in the light of dosage. Indeed, I prefer to see some information about the mice used in this study in the abstract.*

AUTHOR RESPONSE: Abstract has been edited to better reflect the data and includes now also some information about the mice used in this study.

We are also glad to introduce a new co-author to the manuscript; Sofia Rosendahl who had an important role in producing the new data included in the supplemental.

REVIEWERS' COMMENTS:

Reviewer #1 (Remarks to the Author):

In the proposed revised version of their manuscript, the authors have answered to all my previous queries. Of uttermost importance, they appropriately evaluated the differential activities of glucans according to their solubility by realizing a new set of analyses. Also, they better took into account the quality and the purity of commercial products in the experimental design of the study. In summary, I believe that the manuscript can now be published as such.

Reviewer #2 (Remarks to the Author):

The authors answered all of my questions. I don't have any comments.

Rebuttal letter

REVIEWERS' COMMENTS:

Reviewer #1 (Remarks to the Author):

In the proposed revised version of their manuscript, the authors have answered to all my previous queries. Of uttermost importance, they appropriately evaluated the differential activities of glucans according to their solubility by realizing a new set of analyses. Also, they better took into account the quality and the purity of commercial products in the experimental design of the study. In summary, I believe that the manuscript can now be published as such.

Reviewer #2 (Remarks to the Author):

The authors answered all of my questions. I don't have any comments.

Authors comments:

We would like to extend our thanks to the reviewers for their work and the suggestions which have made the article better. We are glad that they appreciate the work we put in and think the manuscript is ready for publication.

As per request in the checklist the following changes to the manuscript has been made:

- 1) The funding agencies have been moved to Acknowledgement
- 2) The text "As a reference anti-CD16/CD32 was diluted 1:80 prior to addition of 15 μ l to cell pellet, after 5min the other antibodies are diluted 1:100 and 15 μ l added to each well and incubated for 20min on ice (no wash in between)." Have been added to the paragraph "Flow cytometry".
- 3) Suppl. Fig. have been changes to Supplementary Figure.
- 4) All supplementary figures have been moved to a PDF.